# Assessment of the Specimen Size Effect on the Fracture Energy of Macro-Synthetic-Fiber-Reinforced Concrete

**DOI:** 10.3390/ma16020673

**Published:** 2023-01-10

**Authors:** Mohammad Daneshfar, Abolfazl Hassani, Mohammad Reza Mohammad Aliha, Tomasz Sadowski

**Affiliations:** 1School of Civil and Environmental Engineering, Tarbiat Modares University, Tehran 14115-111, Iran; 2Welding and Joining Research Center, School of Industrial Engineering, Iran University of Science and Technology (IUST), Narmak, 16846-13114, Tehran, Iran; 3Department of Solid Mechanics, The Lublin University of Technology, Nadbystrzycka 40 Str., 20-2618 Lublin, Poland

**Keywords:** concrete, twisted fiber, fracture energy, mode I fracture toughness, size effect

## Abstract

The most frequently used construction material in buildings is concrete exhibiting a brittle behaviour. Adding fibers to concrete can improve its ductility and mechanical properties. To this end, a laboratory study was conducted to present an experimental model for the specimens’ size effect of on macro-synthetic fiber-reinforced concrete using variations in fracture energy. Composite concrete beams with different thicknesses and widths were made and tested under mode I to obtain (1) fracture toughness, (2) fracture energy, and (3) critical stress intensity factor values. Results indicated that by increasing the thickness and the width, fracture toughness and fracture energy were enhanced. Moreover, increasing the thickness and width of the beam led to critical stress intensity factors enhancement respectively by 35.01–41.43% and 7.77–8.09%.

## 1. Introduction

Fiber-reinforced concrete is a type of composite material made of concrete mixed with fibers of various types and sizes, which can additionally include glass, polymer, carbon, and steel [1].

Studies have been conducted to investigate the effect of fibers on concrete, such as its mechanical properties, fatigue life, and durability, which indicate the positive effect of fibers on concrete performance [2,3,4,5]. Ahmad ([6]) indicated that polypropylene fiber improves the mechanical strength and durability of concrete (particularly tensile capacity) but decreases the flowability of concrete. The optimum dose is important, as a higher dose adversely affects strength and durability due to a lack of flowability. Scanning electronic microscopy results indicate that polypropylene fibers restrict the propagation of cracks, which improves the strength and durability of concrete [6]. Wang et al. reported that with the increase in the steel fiber volume fraction, some fracture parameters increase gradually and maintain a certain linear growth [7].

Pakravan et al. focused on the use of hybrid fibers in concrete and reported that combining various types of fibers would yield better results in terms of concrete toughness and energy absorption [8]. Bordelon and Roesler studied fiber-reinforced concrete pavements using steel, synthetic, and steel mesh fibers. They reported that the use of fibers leads to increased bearing capacity and reduces the thickness of the concrete pavement [9]. Chari et al. investigated the mode I fracture behavior of high-strength steel-fiber-reinforced concrete, the results of which indicated that increased beam size leads to enhanced fracture energy [10]. The shear stress of three beams with different sizes was analyzed by Gustafsson et al. [11]. Their results showed that a concrete mixture containing steel fibers yields better strength. Kreiger [12] conducted a study on a model to explain the mode I rupture of high-performance steel-fiber-reinforced concrete and concluded that increasing the fiber percentage leads to higher fracture energy. In addition, by increasing the span-length-to-depth ratio of the beam, the maximum rupture force significantly reduced. Rao and Rao conducted a study on the toughness change in steel-fiber-reinforced concrete, estimated by mode II loading. Adding fibers to the concrete significantly influenced the concrete’s toughness and shear strength [13]. Different methods were used to rank the toughness of fiber-reinforced concrete, toughness optimization, and the properties of the reinforced concrete.

A summary of recent studies on the effects of different types of fibers on concrete’s mechanical properties is presented in Table 1.

There are numerous studies on the mechanical properties of fiber-reinforced concrete and the size effect of concrete specimens. However, most of the studies related to the size effect increased the thickness and span of the beam specimens. This approach does not describe the effect of thickness or width variations independently.

In this study, we evaluated the effect of the size of a macro-synthetic-fiber-reinforced concrete specimen on the variations in fracture energy. The studied parameters were the fracture energy of notched concrete beams and the stress intensity factor. We used twisted fibers, which were added to the concrete mix with a volume fraction of 0.4%. Three samples were developed for each specimen, and the results were averaged and recorded in tables.

This research was conducted to achieve the following goals:The experimental model of variations in concrete fracture energy considers thickness, width, and macro-synthetic fiber content.The effect of specimen size on the fracture energy of concrete specimens is explained.The effect of fibers on the stress intensity factor of concrete specimens with three different thicknesses and widths is described.

## 2. Materials and Methods

### 2.1. Test Variables

To evaluate the fracture energy and stress intensity factor of notched concrete beams, a concrete mixture was designed based on the ACI 211 standard [29]. All the concrete samples were developed with the same mix design and 0, 0.4, and 0.6 volume percentages of twisted fibers. In this research, concrete mix designs were coarse aggregate 880 (kg/m^3^), fine aggregate 789 (kg/m^3^), cement 442 (kg/m^3^), water 199 (kg/m^3^), superplasticizer 2.2 (kg/m^3^) m^3^), and fibers with two doses of 3.6 and 5.4 (kg/m^3^). The fibers used in this research were selected according to the specifications of ASTM D7508/D7508M-10, with a length of 3.8 cm [30]. Figure 1 shows the fibers used in this research.

### 2.2. Specimen Preparation

To evaluate the fracture energy and critical stress intensity factor (*K*_Ic_) of fiber-reinforced concrete, concrete beam samples were fabricated based on the JCI-S-001-2003 standard [31]. First, cement was mixed with sand, gravel, and fibers; then water was mixed with a superplasticizer accordingly; and finally, several rectangular beam specimens with and without twisted fibers were manufactured. Table 2 shows the geometrical properties of the prepared specimens.

### 2.3. Fracture Energy Tests

Fracture energy is defined as the amount of energy required for crack growth per unit area along the ligament. The fracture energy of the manufactured concrete specimens was measured based on the Japan Concrete Institute’s standard [31]. In this research, the geometry of specimens was as shown in Figure 2. Next, using a three-point bend test, the diagram of load–crack mouth opening displacement (CMOD) was obtained. The area under the curve was measured, and fracture energy *G*_F_ was calculated using Equations (1) and (2).
(1)$$G_{F}\;=\;\frac{0.75\;W_{0}\;+\;W_{1}}{A_{\mathrm{lig}}}$$
(2)$$W_{1}\;=\;0.75\;(\frac{S}{L}m_{1}\;+\;2m_{2})\;g\;{\mathrm{CMOD}}_{C}$$where

*G*_F_—the fracture energy (N/mm^2^),*W*_0_—the area below the CMOD curve up to rupture of the specimen (Nmm),*W*_1_—the work done by the deadweight of the specimen and loading jig (a piece between the testing machine and the specimen; Nmm),*A*_lig_—the area of the broken ligament (b × h; mm^2^),*m*_1_—the mass of the specimen (kg),*S*—the loading span (mm),*L*—the total length of the specimen (mm),*m*_2_—the mass of the jig not attached to the testing machine but placed on the specimen.

The adjustment of the device and specimen is presented in Figure 3.

### 2.4. Critical Stress Intensity Factor

As an important parameter in fracture mechanics, the critical value of the stress intensity factor shows the resistance of a material to crack growth. When a cracked specimen is exposed to remote load, high-stress intensity occurs around the tip of the crack, and when this stress reaches its critical value, the fracture process is initiated in the specimen. The value of the stress intensity coefficient, which is calculated based on this critical stress, is known as the critical stress intensity factor (*K*_Ic_) [31,32]. The stress intensity factor $k_{I}$ for a notched beam subjected to three-point bend loading is obtained from Equation (3).
(3)$$k_{I}\;=\;\frac{PS}{TH^{1.5}}\left[2.9{\left(\frac{a}{H}\right)}^{0.5}\;-\;4.6{\left(\frac{a}{H}\right)}^{1.5}\;+\;21.8{\left(\frac{a}{H}\right)}^{2.5}\;-\;37.6{\left(\frac{a}{H}\right)}^{3.5}\;+\;38.7{\left(\frac{a}{H}\right)}^{4.5}\right]$$where *P*, *H*, *T*, and *S* are shown in Figure 2.

## 3. Results and Discussion

### 3.1. Fracture Energy

To calculate the fracture energy of fiber-reinforced concrete specimens, notched specimens with three different thicknesses (8, 10, and 15 cm) and widths (5, 10, and 15 cm) were manufactured based on the JCI-S-001-2003 standard [31] and loaded by three-point bend loading. The P-CMOD diagrams were obtained from the experiments, and the fracture energy of each concrete specimen was calculated by measuring the area under the curve using Equations (1) and (2). Figure 4 depicts an example of load–CMOD curves of the tested notched fiber-reinforced concrete specimens with a 0.4% fiber volume fraction with different thicknesses.

To study the effect of independent variables, including the thickness of concrete specimens, and fiber dosage on fracture energy, multiple linear regressions were used. The initial equation considered for analysis is shown in Equation (4).
(4)$$GF\;=\;\beta_{1}H\;+\;\beta_{2}Fiberdosage\;+\;\beta_{3}H^{2}\;+\;\beta_{4}Fiberdosage^{2}\;+\;\beta_{5}H.Fiberdosage$$where:

*G*_F_—fracture energy (N/mm^2^)*H*—thickness of concrete specimens (CM)*Fiberdosage*—fiber volume fraction (%)
$\beta_{1},\beta_{2},\beta_{3},\beta_{4},\beta_{5}$ are regression coefficients.

After fitting, coefficients of their significance values were less than the present significance level and were eliminated by the stepwise regression and backward elimination method, and the final model was obtained according to Table 3 and Table 4. Following Table 4, the quadratic term for thickness (*H*) and the linear term for fiber dosage was meaningful if the *p*-values were less than 0.05 with 95% reliability. In the final model, the interactive term was eliminated due to insignificance, which means that the thickness (*H*) and percentage of the fibers do not interact with each other. According to Table 3, the high value of F indicates the overall significance of the model at a high confidence level. However, the adjusted *R*^2^ value of 0.98 indicates that two variables, thickness (*H*) and fiber dosage, explain about 98% of the fracture energy variations; hence, the model is suitable to predict fracture energy.

As shown in Table 4, the coefficient H^2^ was equal to 0.00122. Therefore, if the change in fracture energy for thickness (*H*), holding other factors fixed, is considered, the equation ∆GF = 2 × 0.00122 × H × ∆H would show the fracture energy change value by thickness change (which is plotted for changes from 1 to 6 cm in Figure 5). These diagrams can be used for optimal design and economic evaluation. However, considering fiber dosage coefficients, each 1% increase in the amount of fiber increases the fracture energy by 0.09. To investigate the effect of two variables, thickness and fiber dosage, on fracture energy, considering that the units of these two variables are not the same, the effects of these variables cannot be compared with each other according to their coefficients. For this purpose, the coefficients must first be standardized so that it is possible to compare. The values of these standard coefficients are presented in the last column of Table 4. As can be seen, the higher the value of the coefficient H^2^, the greater the effect of the thickness variable (*H*) relative to the fiber dosage on the fracture energy.

According to the model, the relationship between the three mentioned variables is graphically plotted in Figure 6, which is parabolic. As shown in Figure 6, the thickness and fiber dosage had a positive effect on the fracture energy, and by increasing each, the fracture energy increased, and the highest fracture energy occurred at the highest values of thickness and fiber dosage.

Figure 7 shows an example of the load–CMOD curve of the notched fiber-reinforced concrete specimens with different widths and a 0.4% volume fraction. 

To study the effect of independent variables, including the width of concrete specimens and fiber dosage, on fracture energy, multiple linear regressions were used. The initial equation considered for analysis is shown in Equation (4) by replacing *T* with *H*.

Similar to the method used for thickness variations, a width change was also made and the final model was obtained according to Table 5 and Table 6. According to Table 5, the high value of *F* indicates the overall significance of the model at a high confidence level. However, the adjusted *R*^2^ value of 0.98 indicates that two variables, width (*T*) and *Fiberdosage*, explain about 96% of the fracture energy variations; hence, the model is suitable to predict fracture energy.

As shown in Table 6, the coefficient T^2^ was equal to 0.00051. Therefore, if the change in fracture energy for width (*T*), holding other factors fixed, is considered, the equation ∆GF = 2 × 0.00051 × T × ∆T would show the fracture energy change value by width change (which is plotted for changes from 1 to 6 cm in Figure 8). These diagrams can be used for optimal design and economic evaluation. However, considering fiber dosage coefficients, each 1% increase in the amount of fiber increases the fracture energy by 0.26. To investigate the effect of two variables, width and fiber dosage, on fracture energy, considering that the units of these two variables are not the same, the effects of these variables cannot be compared with each other according to their coefficients. For this purpose, the coefficients must first be standardized so that it is possible to compare. The values of these standard coefficients are presented in the last column of Table 6. As can be seen, the higher the value of the fiber dosage coefficient, the greater effect of the fiber dosage relative to the width variable (*T*) on the fracture energy.

According to the model, the relationship between the three mentioned variables is graphically plotted in Figure 9, which is parabolic. The width and fiber dosage had a positive effect on the fracture energy, and by increasing each, the fracture energy increased, and the highest fracture energy occurred at the highest values of width and fiber dosage.

### 3.2. Critical Stress Intensity Factor

The critical stress intensity factors of different notched macro-synthetic fiber-reinforced concrete specimens were calculated based on Equation (3) and are presented in Table 7 and Table 8.

As seen from these results, generally, the value of *K*_Ic_ increases by increasing the thickness *H* and width *T* of concrete beams.

### 3.3. Assessment of Ruptured Cross Section and Fibers

After the tests, the cross-sectional area of the sample was broken and the tip of the fiber was examined.

Figure 10 shows the number of broken sections of the fiber. A cross-sectional analysis of the broken sample showed that most of the sample failure was from aggregates and the mixture design was suitable. In addition, according to the figure, the tip of the fiber shows that the fibers did not rupture due to elongation and were not pulled out, which indicates good performance of the fibers.

## 4. Conclusions

In this study, we calculated the fracture energy for macro-synthetic fiber-reinforced concrete specimens with different thicknesses and widths tested using notched beam specimens. The main concluding remarks are:The experimental model of the effect of specimen size was presented by testing specimens with three different thicknesses and widths for normal and fiber-reinforced concrete.The results in Table 4 and Table 6 show the relationship between the fracture energy and the thickness and width of the parabola.The results in Table 4 and Figure 5 and Figure 6 indicate that adding fibers and increasing the thickness will increase the fracture energy.The results in of Table 6 and Figure 8 and Figure 9 indicate that adding fibers and increasing the width will increase the fracture energy.The results in Table 4 and Table 6 show that by assuming constant fracture energy, the thickness of fiber-reinforced concrete samples is lower than that of conventional concrete, which can be considered in the design of concrete structures, especially concrete pavement.The results in Table 7 indicate that adding fibers and increasing the thickness will increase the critical stress intensity factor up to 52.14%.The results in Table 8 indicate that adding fibers and increasing the width will increase the critical stress intensity factor up to 13.05%.

## Figures and Tables

**Figure 1 materials-16-00673-f001:**
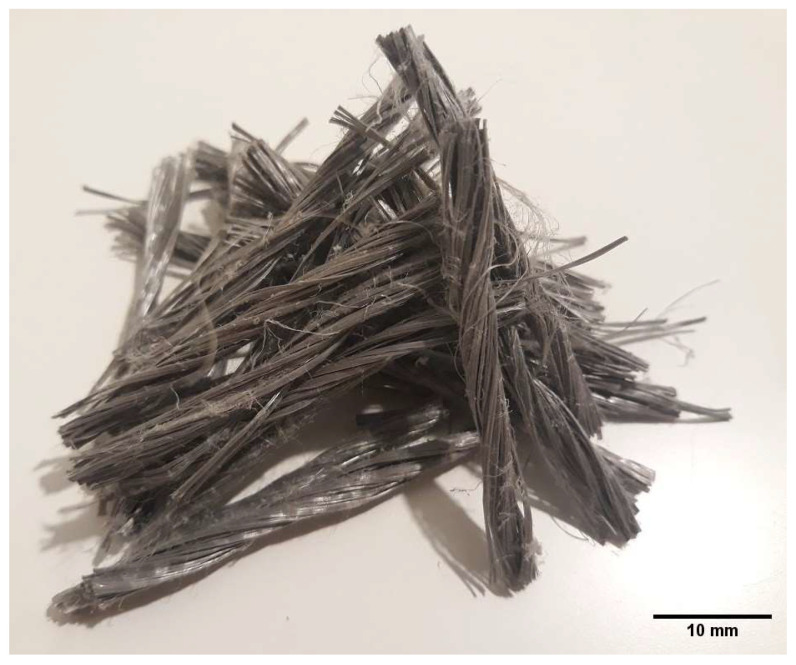
Twisted fibers used in the manufacturing of fiber-reinforced concrete.

**Figure 2 materials-16-00673-f002:**
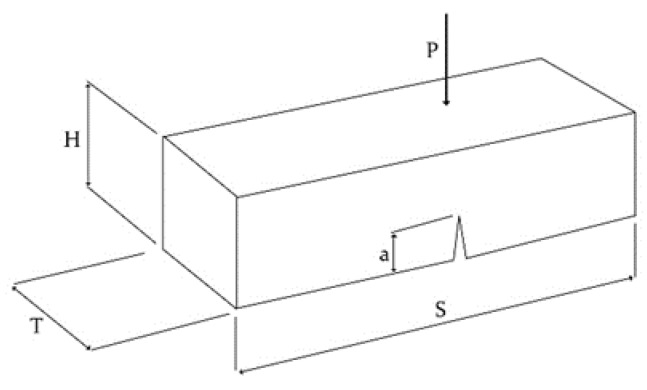
The geometry of the tested specimen [28].

**Figure 3 materials-16-00673-f003:**
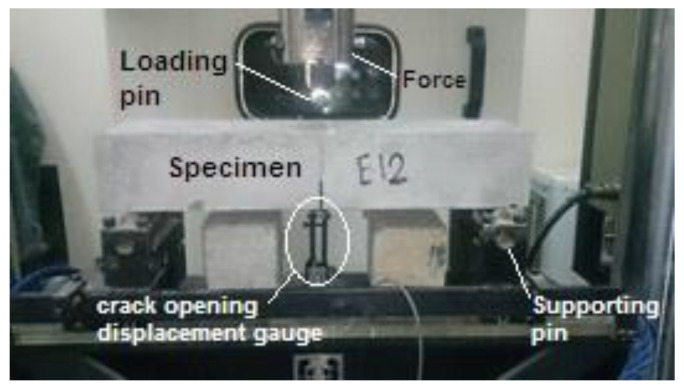
Adjustment of the device and specimen to measure the load–CMOD.

**Figure 4 materials-16-00673-f004:**
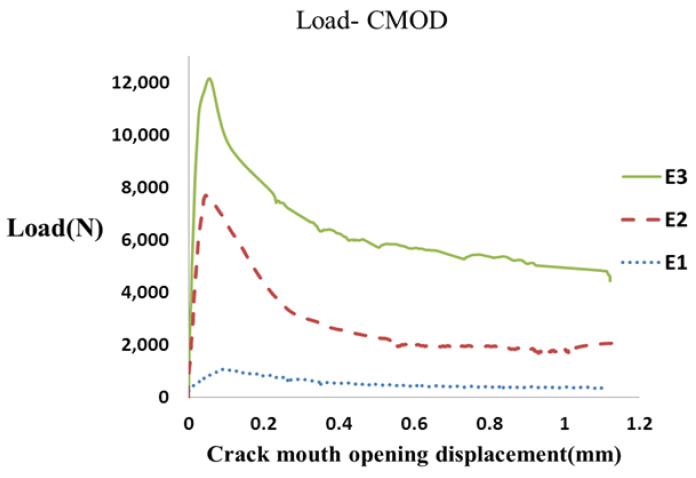
The load–CMOD curve obtained for fiber-reinforced concrete specimens with different thicknesses.

**Figure 5 materials-16-00673-f005:**
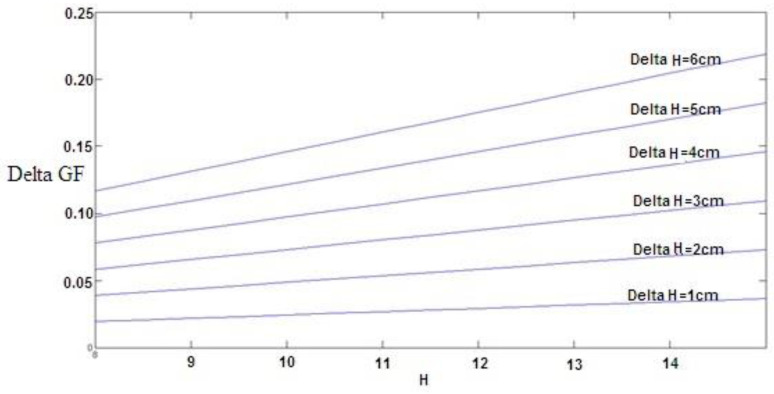
The effect of thickness *H* change on fracture energy *G*_F_.

**Figure 6 materials-16-00673-f006:**
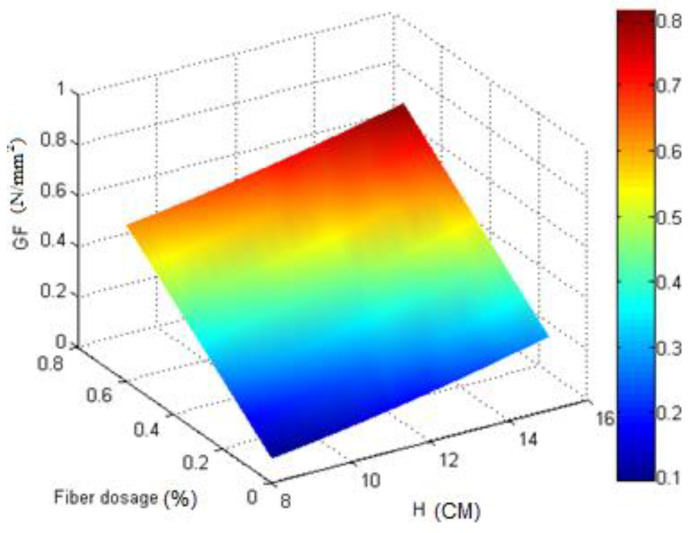
The relationship between *G*_F_, *H*, and *Fiberdosage*.

**Figure 7 materials-16-00673-f007:**
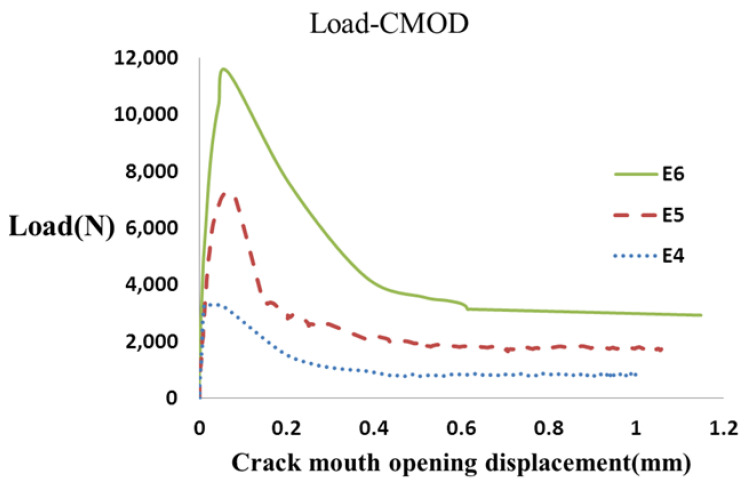
The load–CMOD curve of fiber-reinforced concrete specimens with different widths *T*.

**Figure 8 materials-16-00673-f008:**
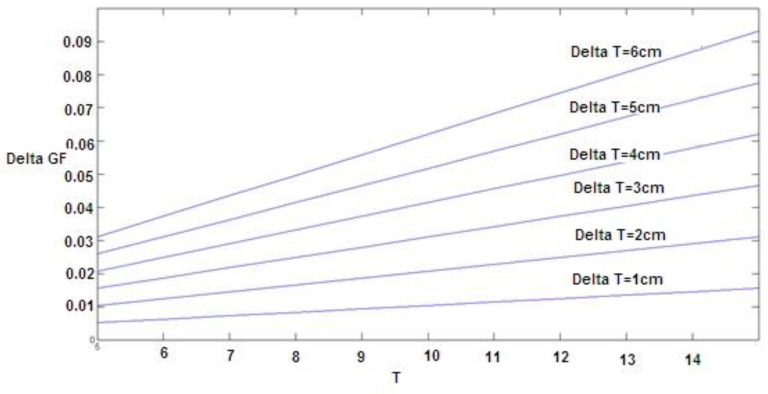
The effect of width change on fracture energy.

**Figure 9 materials-16-00673-f009:**
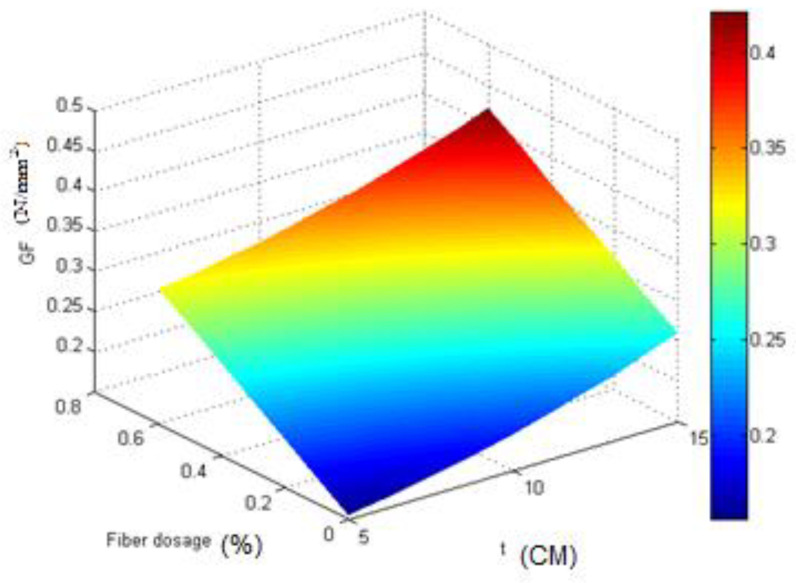
The relationship between *G*_F_, *T*, and *Fiberdosage*.

**Figure 10 materials-16-00673-f010:**
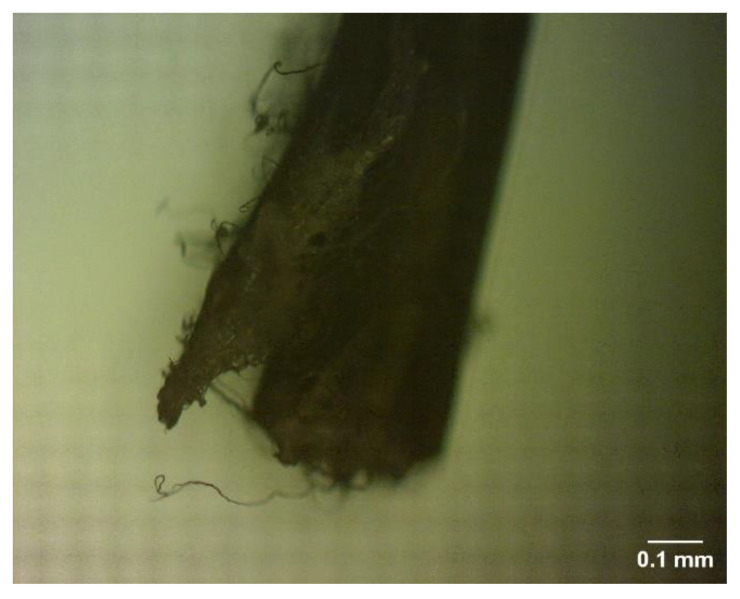
The cross-sectional area of a fiber.

**Table 1 materials-16-00673-t001:** A summary of fibers’ effects on mechanical properties, as reported in the literature (increase ↑, decrease ↓, not significant, N.S.).

Authors	Year	Fiber Properties			Concrete Type	Compressive Strength	Splitting Tensile Strength	Flexural Strength	Energy Absorption
		Type	Length (mm)	Fiber Volume Fraction (%)					
Valdez et al. [14]	2021	Steel fibers	50	0.25, 0.5	Normal concrete	-	-	-	↑
Blazy et al. [15]	2021	Polypropylene	48, 54	0.22, 0.33	Normal concrete	-	-	5.55–13.5% ↑	-
Daneshfar et al. [16]	2017	Polypropylene	38	0.2, 0.4, 0.6	Normal concrete	4.57–26.32% ↓	0.84–34.29% ↑	19.6–81.69% ↑	-
Fallah and Nematzadeh [17]	2017	Polypropylene	39	0.25, 0.75, 1.25	High-strength concrete	8% ↑, 3% ↑, 4% ↓	8, 9, 27% ↑	-	-
Lee et al. [18]	2017	Steel fibers	20, 30, 40	0.25, 0.375, 0.5	Normal concrete	-	-	At least 20.8% ↑	↑
Alberti et al. [19]	2017	Steel fibers (hooked)	35	0.33	Self-compacting concrete	↓	↑	↑	↑
		Polypropylene	60	0.5					
Hesami et al. [20]	2016	Polypropylene	60	0.10, 0.12	Self-compacting concrete	2% ↑, 5% ↓	19, 27% ↑	26, 33%	-
Saidani et al. [21]	2016	Steel fibers	50	4% (by cement volume)	Normal concrete	2% ↓	98% ↑	-	-
		Polypropylene	50	4% (by cement volume)		5% ↓	65%↑	-	-
Afroughsabet and Ozbakkaloglu [22]	2015	Hooked-end steel	60	0.25, 0.5, 0.45, 1	High-strength concrete	12, 14, 15, 19% ↑	15, 22, 38, 57% ↑	14, 28, 36, 61% ↑	-
Yew et al. [23]	2015	Polypropylene (twisted bundle)	54	0.25, 0.375, 0.5	Lightweight concrete	5, 11, 15% ↑	8, 24, 33% ↑	29, 31, 40%↑	-
			30	0.25, 0.375, 0.5		3, 10, 14% ↑	10, 19, 27% ↑	18, 22, 30%↑	-
		Polypropylene (straight)	20	0.25, 0.375, 0.5		4, 10, 14% ↑	13, 14, 21%↑	6, 10, 20%↑	-
Karadelis and Yougui [24]	2015	Steel	50	1.5	Roller-compacted concrete	N.S	-	24%↑	-
Hesami et al. [25]	2014	Steel	36	0.5	Previous concrete	24% ↑	33%↑	19%↑	-
		PPS	54	0.3		28% ↑	37%↑	21%↑	-
		Glass	12	0.2		32% ↑	28%↑	17%↑	-
Pajak and Ponikiewski [26]	2013	Hooked-end steel	30	0.5, 1, 1.5	Self-compacting concrete	34, 32, 20% ↑	-	55, 151, 339% ↑	-
Singh et al. [27]	2010	Steel fibers (corrugated)	35	1	Normal concrete	18% ↑	-	80%↑	↑
		Polypropylene (fibrillated)	60						
Silva and Thaumaturgo [28]	2002	Wollastonite	20	2, 3, 5	Geopolymer concrete	-	-	-	80% ↑

**Table 2 materials-16-00673-t002:** Specifications of tested specimens.

Specimen No.	Shape of Fiber	Fiber Volume Fraction (%)	Specimen Size (mm)	Notch Length (mm)	Notch Width (mm)
E1	Twisted	0.4, 0.6	80 × 120 × 450	30	2
E2	Twisted	0.4, 0.6	100 × 120 × 450	30	2
E3	Twisted	0.4, 0.6	150 × 120 × 450	30	2
E4	Twisted	0.4, 0.6	100 × 50 × 350	30	2
E5	Twisted	0.4, 0.6	100 × 100 × 350	30	2
E6	Twisted	0.4, 0.6	100 × 150 × 350	30	2
N1	-	0	80 × 120 × 450	30	2
N2	-	0	100 × 120 × 450	30	2
N3	-	0	150 × 120 × 450	30	2
N4	-	0	100 × 50 × 350	30	2
N5	-	0	100 × 100 × 350	30	2
N6	-	0	100 × 150 × 350	30	2

**Table 3 materials-16-00673-t003:** ANOVA table.

	DF	SS	MS	F-Value	*p*-Value
Total	8	0.06862	0.008578	-	-
Model	2	0.06757	0.033786	192.98	0.0000
Residual error	6	0.00105	0.000175	-	-
R^2^	0.985	-	-	-	-
Adjusted R^2^	0.980	-	-	-	-

**Table 4 materials-16-00673-t004:** Values of regression coefficients of G_F_.

Independent Variable	Regression Coefficient	*p*-Value	*t*-Value	Standard Error	Standardized Coefficients
Constant	0.01781	0.159	1.61	0.0000639	-
H^2^	0.00122	0	19.02	0.0012159	0.9607599
Fiber dosage	0.0869	0.003	4.92	0.011086	0.2482591

**Table 5 materials-16-00673-t005:** ANOVA table.

Total	DF	SS	MS	F-Value	*p*-Value
Model	8	0.0584	0.007302	-	-
Residual error	2	0.0569	0.028428	109.12	0
R^2^	6	0.0016	0.00026		
Adjusted R^2^	0.9732	-	-	-	-
	0.9643	-	-	-	-

**Table 6 materials-16-00673-t006:** Values of regression coefficients of *G*_F_.

Independent Variable	Regression Coefficient	*p*-Value	*t*-Value	Standard Error	Standardized Coefficients
Constant	0.1438929	0	12.23	0.0117701	-
T2	0.0005169	0	7.93	0.0003574	0.5293087
Fiber dosage	0.2588928	0	12.47	0.2161148	0.8325113

**Table 7 materials-16-00673-t007:** Critical stress intensity factor for different thicknesses *H*.

Specimen	PAVE (N)	a (mm)	H (mm)	a/H	T (mm)	K (MPa × √m)	Percentage Change w.r.t Plane Concrete (%)
N1	2080	30	80	0.375	120	17.48972	0
N2	5082.9	30	100	0.3	120	25.15605	0
N3	13,211.2	30	150	0.2	120	27.3126	0
E1-0.4	2880.1	30	80	0.375	120	24.21737	38.46
E2-0.4	6701.5	30	100	0.3	120	33.16676	31.84
E3-0.4	16,821.1	30	150	0.2	120	34.77565	27.32
E1-0.6	3164.6	30	80	0.375	120	26.60959	52.14
E2-0.6	7704.7	30	100	0.3	120	38.13175	51.58
E3-0.6	19,099.6	30	150	0.2	120	39.48618	44.57

**Table 8 materials-16-00673-t008:** Critical stress intensity factor for different widths *T*.

Specimen	PAVE (N)	a (mm)	H(mm)	a/H	T (mm)	K (MPa × √m)	Percentage Change w.r.t Plane Concrete (%)
N4	3318.9	30	100	0.3	50	30.32	0
N5	7240.6	30	100	0.3	100	33.08	0
N6	11,505.8	30	100	0.3	150	35.04	0
E4-0.4	3587.3	30	100	0.3	50	32.78	8.09
E5-0.4	7774.1	30	100	0.3	100	35.52	7.37
E6-0.4	12,400	30	100	0.3	150	37.77	7.77
E4-0.6	3752.1	30	100	0.3	50	34.28	13.05
E5-0.6	8101.4	30	100	0.3	100	37.01	11.88
E6-0.6	12,691.8	30	100	0.3	150	38.65	10.3

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
