# Peer review of "Assessment of the Specimen Size Effect on the Fracture Energy of Macro-Synthetic-Fiber-Reinforced Concrete"

_materials, 2023, doi:10.3390/ma16020673_

Round 1

Reviewer 1 Report

Introduction part to be rewritten with proper background of the work in the chronological order.

Content of Table 2 is hidden at the right border.

Figure shows bundled fibers. Is this the one used in the study? If so, how the defined size has been achieved?

Line 73 mentions fibres were added at 0.4% volume fraction, but Table 4 shows 0.4% and 0.6%? Justify.

m2 is not visible in Figure 3. Change Figure 3 with exact load setup.

Line 137 & 138 mentions "notched specimens with three different thicknesses of (8, 10, and 15 cm) and widths of (5, 10, and 15 cm) at least three samples from each specimen were manufactured", but not matched with the size mentioned under Table 3.

"multiple linear regressions", which tool has been used?

R2 of 0.98 is too high, which shows that the equation developed is too accurate. But, analytical studies are not possible to be done with minimum range of values.

What is H2 and T2 from Table 6 & 8 respectively? What is the significance of these?

How can you define the curves in Figure 6 & 9 are parabolic?

Figure 10 shows broken fibre. Is this steel?

What is the overall outcome of this study?

Author Response

Dear Reviewers

Authors would like to thank reviewers for the time they dedicated to review the paper. Here are the responses to the issues raised. The parts that have been added to the text are shown in highlighted color.

Reviewer #1:

Comment: Introduction part to be rewritten with proper background of the work in the chronological order.

Answer: checked and corrected.

Comment: Content of Table 2 is hidden at the right border.

Answer: checked and corrected.

Comment: Figure shows bundled fibers. Is this the one used in the study? If so, how the defined size has been achieved?

Answer: In this research, the fibers available in the market and manufactured by Forta Company were used according to the specifications of Table 3.

Comment: Line 73 mentions fibres were added at 0.4% volume fraction, but Table 4 shows 0.4% and 0.6%? Justify.

Answer: checked and corrected.

Comment: m2 is not visible in Figure 3. Change Figure 3 with exact load setup.

Answer: Because  of the surface of the concrete sample was prepared completely smooth, The piece in question was not used in this test and the weight of the m2 was considered zero.

Comment: Line 137 & 138 mentions "notched specimens with three different thicknesses of (8, 10, and 15 cm) and widths of (5, 10, and 15 cm) at least three samples from each specimen were manufactured", but not matched with the size mentioned under Table 3.

Answer: The dimensions of the sample are listed in Table 4 and Table 3 is the specifications of the fibers.

Comment: "multiple linear regressions", which tool has been used?

Answer: Data analysis was done with Stata software.

Comment: R2 of 0.98 is too high, which shows that the equation developed is too accurate. But, analytical studies are not possible to be done with minimum range of values.

Answer: Overfitting has occurred due to the number of tests.

Comment: What is H2 and T2 from Table 6 & 8 respectively? What is the significance of these?

Answer: The aim of this research was to model the relationship between sample dimensions and fracture energy, and as mentioned in the text, H is the height and T is the width of the sample.

Comment: How can you define the curves in Figure 6 & 9 are parabolic?

Answer: The curves of figures 6 and 9 are the results of the obtained models and show the correlation of the fracture energy with the thickness and width of the samples in a parabolic form.

Comment: Figure 10 shows broken fibre. Is this steel?

 Answer: According to Table 3, the fibers used in this research are made of polypropylene.

Comment: What is the overall outcome of this study?

Answer: Presenting an empirical model of the relationship between specimen size and fracture energy.

Reviewer 2 Report

The manuscript, titled "Assessment of specimen size effect on fracture energy of macro-synthetic fiber-reinforced concretes," presents an intriguing idea of a study conducted on the effect of wall thickness on fiber-reinforced concrete fracture resistance. However, the paper is limited by a very low number of mixtures and conventional testing methods. The study cannot present the optimal parameters in its current form, and the experimental results and discussions are not available.

Reliable. The paper should be rejected. Some comments follow:

Introduction section

Could the authors please explain the following sentence more clearly: "Fiber-reinforced concrete is a type of composite material that is mixed with fibers? As presented, concrete is a composite material, which isn’t confirmed by the literature. Maybe the authors should present this as "Fiber-reinforced concrete is a type of composite material made of concrete mixed with fibers."

"We utilized twisted fibers, which were added to the concrete mix with a volume fraction of 0.4%"—please check your experimental results. In the materials and methods sections, the authors stated that they used two mixtures (0.4 and 0.6).

Also, what was the rationale behind choosing twisted fibers?

Materials and Methods

 The choice of some parameters has no rationale. Why did the author choose to study only a mix of 0.4% and 0.6% fibers? Why not higher or lower amounts?

Table 2 is incomplete. Please use the journal template. The values cannot be read.

How many fibers are included in one rod (it seems that not all rods have the same number of fibers)?

Figure 1: Please introduce the scale bar on the figure.

This section is highly unclear. Table 2 shows only one mixture, while Table 4 shows three different mixtures...

What was the age of the samples on the testing day?

Results and Discussion

Figure 4 is unclear. Please provide higher-resolution images.

The relationship between GF, H, and fiber dosage This analysis is very interesting; however, it also clearly shows the main limitation of this study, which is the very low number of mixtures. Therefore, an optimum value of the thickness or fiber dosage wasn’t identified. The study is only the beginning of a good research direction.

Ruptured Cross Section and Fibers Evaluation

The images and the description aren’t related. The claims and affirmations presented by the authors can’t be confirmed by the provided results. Please provide SEM analysis or involve techniques and equipment that provide higher magnifications.

Overall.

The idea of the study is interesting. However, this is only the beginning of a research project. The study should include much more mixtures, and the results should be confirmed using microstructural analysis with high magnification.

The study makes numerous claims without providing context for the presented experimental results. The article needs significant revisions and improvements to make it suitable for publication in Materials.

Author Response

Dear Reviewers

Authors would like to thank reviewers for the time they dedicated to review the paper. Here are the responses to the issues raised. The parts that have been added to the text are shown in highlighted colour.

Reviewer 2: The manuscript, titled "Assessment of specimen size effect on fracture energy of macro-synthetic fiber-reinforced concretes," presents an intriguing idea of a study conducted on the effect of wall thickness on fiber-reinforced concrete fracture resistance. However, the paper is limited by a very low number of mixtures and conventional testing methods. The study cannot present the optimal parameters in its current form, and the experimental results and discussions are not available. The idea of the study is interesting. However, this is only the beginning of a research project. The study should include much more mixtures, and the results should be confirmed using microstructural analysis with high magnification. The study makes numerous claims without providing context for the presented experimental results. The article needs significant revisions and improvements to make it suitable for publication in Materials.

Comment: Could the authors please explain the following sentence more clearly: "Fiber-reinforced concrete is a type of composite material that is mixed with fibers? As presented, concrete is a composite material, which isn’t confirmed by the literature. Maybe the authors should present this as "Fiber-reinforced concrete is a type of composite material made of concrete mixed with fibers."

Answer: checked and corrected.

Comment:"We utilized twisted fibers, which were added to the concrete mix with a volume fraction of 0.4%"—please check your experimental results. In the materials and methods sections, the authors stated that they used two mixtures (0.4 and 0.6).

Answer: checked and corrected.

Comment: Also, what was the rationale behind choosing twisted fibers?

Answer: The fibers produced by Forta and Barchip companies are two types of macrosynthetic fibers that are most widely used in the construction industries. According to the authors' previous researches (line 13 of references), the performance of fiber concrete reinforced with two types of fibers is almost the same, that's why in this research, the fibers produced by Forta Company (twisted) were used.

Comment: The choice of some parameters has no rationale. Why did the author choose to study only a mix of 0.4% and 0.6% fibers? Why not higher or lower amounts?

Answer: In this research, based on previous studies, different percentages of fibers were selected, and according to the presentation of the experimental model in this research, choosing a specific amount of fibers has no effect on the presented model.

Comment:Table 2 is incomplete. Please use the journal template. The values cannot be read.

Answer: checked and corrected.

Comment: How many fibers are included in one rod (it seems that not all rods have the same number of fibers)?

Answer: The fibers are manufactured by the factory and have the same specifications.

Comment:Figure 1: Please introduce the scale bar on the figure.

Answer: The physical characteristics of the fibers are shown in Table 3.

Comment:This section is highly unclear. Table 2 shows only one mixture, while Table 4 shows three different mixtures...

Answer: checked and corrected.

Comment:What was the age of the samples on the testing day?

Answer: According to the standard, the age of the tested specimens is 28 days.

The relationship between GF, H, and fiber dosage This analysis is very interesting; however, it also clearly shows the main limitation of this study, which is the very low number of mixtures. Therefore, an optimum value of the thickness or fiber dosage wasn’t identified. The study is only the beginning of a good research direction.

Comment:The images and the description aren’t related. The claims and affirmations presented by the authors can’t be confirmed by the provided results. Please provide SEM analysis or involve techniques and equipment that provide higher magnifications.

 Answer: checked and corrected.

Reviewer 3 Report

The research plan is focused on the area of Assessment of specimen size effect on fracture energy of macro-synthetic-fiber-reinforced concretes.

The research topic is interesting, but on the other hand, many authors are devoted to the addressed area.

The structure of the manuscript is typical, but the experimental program could be more ambitious, both in terms of samples and individual tests.

It would be appropriate to solve the topic using also numerical modeling or to focus on the microstructure and durability of materials.

In the introduction part, it is necessary to better state the motivation and originality of the addressed research area. It is necessary to provide more information about the addressed area, where there is an extensive research program. Recent interesting of ones include of papers , for example: https://doi.org/10.3390/ma15165707  ; https://doi.org/10.3390/fib10030026. Tables in the manuscript are not in template format. Needs to be fixed. The references in the manuscript also do not exactly conform to the MDPI manuscript template.

The description of the experimental tests itself is sufficiently presented.

Using a 3D chart is a nice and interesting data processing.

The basic recapitulation of the results is good, but there is a lack of a broader discussion of the results - pros, cons, limitations, etc. It must be significantly improved. Other possible expansions and improvements include - expansion of the experimental program by more material variants and the inclusion of more types of test experiments.

After editing, the manuscript may be interesting to the journal readers.

Author Response

Dear Reviewers

Authors would like to thank reviewers for the time they dedicated to review the paper. Here are the responses to the issues raised. The parts that have been added to the text are shown in highlighted colour.

Reviewer 3: The research plan is focused on the area of Assessment of specimen size effect on fracture energy of macro-synthetic-fiber-reinforced concretes. The research topic is interesting, but on the other hand, many authors are devoted to the addressed area. The structure of the manuscript is typical, but the experimental program could be more ambitious, both in terms of samples and individual tests. It would be appropriate to solve the topic using also numerical modeling or to focus on the microstructure and durability of materials. The description of the experimental tests itself is sufficiently presented. Using a 3D chart is a nice and interesting data processing. The basic recapitulation of the results is good, but there is a lack of a broader discussion of the results - pros, cons, limitations, etc. It must be significantly improved. Other possible expansions and improvements include – the expansion of the experimental program by more material variants and the inclusion of more types of test experiments. After editing, the manuscript may be interesting to the journal readers.

Comment: In the introduction part, it is necessary to better state the motivation and originality of the addressed research area. It is necessary to provide more information about the addressed area, where there is an extensive research program. Recent interesting of ones include of papers , for example: https://doi.org/10.3390/ma15165707  ; https://doi.org/10.3390/fib10030026.

Answer: checked and corrected.

Comment: Tables in the manuscript are not in template format. Needs to be fixed. The references in the manuscript also do not exactly conform to the MDPI manuscript template.

Answer: checked and corrected.

Round 2

Reviewer 1 Report

Comments are taken care

Author Response

Thank you for review of our manuscript.

We are satisfied with all your suggestions leading to the improve of our manuscript.

Reviewer 2 Report

Dear Authors, 

Thank you for revising your paper according to my suggestions and recommendations. Further, please consider the following suggestions to increase the clarity of your paper:

Please check the conversion or provide higher resolution images if any of the figures are unclear.

Figure 1: Please introduce a scalebar on the figure (it is impossible to determine the dimensions of the fibers without a scalebar).

Figure 3: Please introduce labels on the figure to indicate the areas of interest for the readers.

Figure 10: Please provide a clear image (optical microscopy isn't a suitable technique since you cannot obtain a flat surface). Also, without a scalebar, the readers can't analyze the particularities of the fiber.

 Best regards,

Author Response

Thank you for review of our manuscript.

We are satisfied with all your suggestions leading to the improvement of our manuscript.